# Sequential Treatment with Temozolomide Plus Naturally Derived AT101 as an Alternative Therapeutic Strategy: Insights into Chemoresistance Mechanisms of Surviving Glioblastoma Cells

**DOI:** 10.3390/ijms24109075

**Published:** 2023-05-22

**Authors:** Dana Hellmold, Carolin Kubelt, Tina Daunke, Silje Beckinger, Ottmar Janssen, Margarethe Hauck, Fabian Schütt, Rainer Adelung, Ralph Lucius, Jochen Haag, Susanne Sebens, Michael Synowitz, Janka Held-Feindt

**Affiliations:** 1Department of Neurosurgery, University Medical Center Schleswig-Holstein UKSH, Campus Kiel, 24105 Kiel, Germany; 2Institute of Experimental Cancer Research, University Medical Center Schleswig-Holstein UKSH, Campus Kiel, 24105 Kiel, Germany; 3Institute for Immunology, University Medical Center Schleswig-Holstein UKSH, Campus Kiel, 24105 Kiel, Germany; 4Functional Nanomaterials, Department of Materials Science, Kiel University, 24143 Kiel, Germany; 5Institute of Anatomy, Kiel University, 24098 Kiel, Germany; 6Department of Pathology, Kiel University, 24105 Kiel, Germany

**Keywords:** glioblastoma, R-(-)-gossypol, AT101, temozolomide, chemoresistance, combined therapy, mTOR, stemness, epithelial–mesenchymal transition

## Abstract

Glioblastoma (GBM) is a poorly treatable disease due to the fast development of tumor recurrences and high resistance to chemo- and radiotherapy. To overcome the highly adaptive behavior of GBMs, especially multimodal therapeutic approaches also including natural adjuvants have been investigated. However, despite increased efficiency, some GBM cells are still able to survive these advanced treatment regimens. Given this, the present study evaluates representative chemoresistance mechanisms of surviving human GBM primary cells in a complex in vitro co-culture model upon sequential application of temozolomide (TMZ) combined with AT101, the R(-) enantiomer of the naturally occurring cottonseed-derived gossypol. Treatment with TMZ+AT101/AT101, although highly efficient, yielded a predominance of phosphatidylserine-positive GBM cells over time. Analysis of the intracellular effects revealed phosphorylation of AKT, mTOR, and GSK3ß, resulting in the induction of various pro-tumorigenic genes in surviving GBM cells. A Torin2-mediated mTOR inhibition combined with TMZ+AT101/AT101 partly counteracted the observed TMZ+AT101/AT101-associated effects. Interestingly, treatment with TMZ+AT101/AT101 concomitantly changed the amount and composition of extracellular vesicles released from surviving GBM cells. Taken together, our analyses revealed that even when chemotherapeutic agents with different effector mechanisms are combined, a variety of chemoresistance mechanisms of surviving GBM cells must be taken into account.

## 1. Introduction

Glioblastomas (GBM) are highly malignant brain tumors that are characterized by rapid and infiltrative growth. Despite significant improvements in (micro)surgery and aggressive treatment, the prognosis of patients is poor due to the fast development of tumor recurrences and high resistance to chemo- and radiotherapy [1,2]. Several molecular mechanisms of therapy resistance in GBM are known, e.g., a marked intra- and inter-tumor heterogeneity related to at least three known clinically relevant GBM subtypes (classical, mesenchymal, proneural [3]) and a pro-tumorigenic microenvironment consisting of, e.g., non-neoplastic tumor-associated macrophages and astrocytes [4,5]. Furthermore, the coexistence of multiple tumor subclones including glioma stem-like cells (GSCs), which are individually attributable to defined niches within the tumor, may result in tumor recurrence in part due to the presence of a therapy-resistant subpopulation of GSCs or by the induction of dedifferentiation in the non-GSC subpopulation of GBM cells as a response to the treatment [6]. In addition, alterations or redundancies of signaling pathways are of great importance. For example, the PI3K/Akt/mTOR (phosphatidylinositol 3 kinase/Akt/mechanistic target of rapamycin) pathway is activated in approximately 30% of GBMs, and an elevated expression of the GSK3β (glycogen synthase kinase 3β) is common [7]. Further, a high-level genomic amplification (~40%) occurs in the epidermal growth factor receptor (EGFR) gene, often along with constitutively activating mutations in the EGFR ectodomain mainly through the variant III deletion event (EGFRvIII) [8]. Beyond that, the canonical/noncanonical Wnt/β-catenin pathway plays a key role in the development of GBMs [4,5]. Moreover, the epigenetic repression of DNA repair enzymes due to MGMT (O^6^-methylguanine DNA methyltransferase) promoter hypermethylation or acquired DNA mismatch repair deficiencies, an aberrant cell cycle progression including multiple mutations of regulatory genes, the dysregulations of miRNAs or the induction of pro-tumorigenic processes including elevated epithelial–mesenchymal transition (EMT) or angiogenesis and many more, are important factors for GBM resistance [4,9,10]. In addition, as has recently been reported, GBM cells are able to drive malignant progression via a distinct cellular network and the exchange of extracellular vesicles (EVs) [11]. 

Due to the various factors known to drive chemoresistance in GBMs, the efficacy of chemotherapeutic drugs used in the clinic is limited. This also accounts for temozolomide (TMZ) as the chemotherapeutic drug of choice. Its therapeutic benefit depends on its ability to alkylate/methylate DNA, most commonly at N-7 or O-6 positions of guanine residues [12]. Nonetheless, over 50% of GBM patients treated with TMZ do not respond to the therapy, demonstrating that the treatment regimen is highly dependent on the patient’s history and molecular tumor characteristics [4]. However, in the case of MGMT promoter-methylated GBMs, TMZ is in principle able to control tumor progression and is indeed beneficial for some patients [13], but still, several GBM cells are able to survive.

To overcome the highly adaptive behavior of GBMs, various potential therapeutics for GBMs have been investigated [14,15], also including multimodal therapeutic approaches [16]. Along this line, we recently analyzed the efficacy of a sequentially applied combined TMZ plus AT101 treatment approach to improve the therapeutic efficacy, especially of MGMT promoter-methylated GBMs [17,18,19]. Here, the BH3 mimetic AT101, an R-(-)-enantiomer of the naturally occurring cottonseed-derived polyphenol gossypol [20,21], competitively binds to hydrophobic surface grooves of pro-survival Bcl-2 family members, counteracting their protective effects and facilitating apoptosis [21] as well as autophagic cell death in tumor cells [22,23,24]. Due to their complementing modes of action, the combination of TMZ and BH3 mimetics could be a promising approach for GBM treatment. Indeed, stimulation with TMZ and AT101 resulted in higher cytotoxicity and better tumor growth control of MGMT promoter-methylated, patient-derived, primary GBM cells compared to a single TMZ treatment [17]. This was particularly effective when initial combined treatment with TMZ+AT101 was followed by a single AT101 application [18]. Importantly, this effect was also observed when using dual co-culture conditions with astrocytes and microglial cells in addition to GBM cells. Under these conditions, we noted that this treatment regimen preferentially affected glioma cells, whereas it was less detrimental to healthy brain cells [19]. However, we also faced the fact that some of the GBM cells were still able to overcome this combination treatment, regardless of whether patient-derived primary material or GBM cell lines were studied [18,19]. This raised the question of which intracellular mechanisms might be relevant for overcoming cell death upon TMZ+AT101/AT101 treatment, focusing on chemoresistance as a substantial and clinically meaningful hurdle that must be overcome to successfully treat especially MGMT promoter-methylated GBMs. Therefore, we set up a new series of experiments using live cell imaging to investigate the time course of cell death upon TMZ+AT101/AT101 administration in human primary GBM cells and analyzed the intracellular signaling and the resulting regulation of relevant pro-tumorigenic genes upon therapy. Furthermore, we separated and initially characterized EVs from treated GBM cells. All experiments were performed on mono-cultured, patient-derived primary differentiated GBM cells as well as on GBM cells co-cultured with astrocytes and microglia in a dual indirect co-culture system. Moreover, the efficiency of the TMZ+AT101/AT101 treatment plan was also proven in patient-derived primary GSCs. Of note, we used only GBM cultures with methylated MGMT promoters. Further, in our experimental design, we focused only on resistance mechanisms induced by chemotherapeutic agents, without considering other multimodal therapies, such as concurrent chemotherapy and radiotherapy, and without analyzing the effects on MGMT promoter-unmethylated GBM cells.

## 2. Results

### 2.1. TMZ+AT101/AT101 Treatment Led to an Increased Number of Surviving Phosphatidylserine-Positive GBM Cells over Time

For the primary human GBM cultures used in this work (PCa, PCb), we reproduced the previously reported cytotoxic effect of sequential combination therapy of TMZ+AT101/AT101 in mono-cultured GBM cells as well as in GBM cells co-cultured with astrocytes and microglia for a total cultivation period of 6 days. Importantly, the used primary cultures were both characterized by a methylated MGMT promoter (PCa: 91.8%; PCb: 84.2%). Regarding the co-cultivation conditions, in accordance with the results published by our group before [19], we chose cell ratios reflecting the condition after an incomplete resection of the tumor, namely 70% GBM cells, 29% microglia, and 1% astrocytes. In agreement with our previous report, there was a prominent cytotoxic and antiproliferative effect of the treatment after 6 days, although this was somewhat more pronounced in the mono-culture. Overall, a maximum of approximately 75% of GBM cells died after 6 days of therapy (Figure 1A,D). 

To gain a better insight into the dynamics of GBM cell death, we performed live cell imaging using the Polarity Sensitive Indicator of Viability pSIVA^(TM)^ system throughout the treatment period in both mono- and co-cultured GBM cells. This is an annexin-based, polarity-sensitive probe for spatiotemporal or kinetic analysis of apoptosis and other forms of cell death. The binding of pSIVA^(TM)^ is reversible, allowing the detection of transient phosphatidylserine (PS) exposure to the external leaflet of the plasma membrane, which is associated with normal physiological processes as well as reversible or recoverable apoptotic cell death events. pSIVA^(TM)^ is conjugated to IANBD, a polarity-sensitive dye that fluoresces green only when pSIVA^(TM)^ is bound to the cell membrane, while late-apoptotic cells show red fluorescence due to irreversible uptake of propidium iodide (PI).

For both PCa and PCb, the signal intensities of green and red fluorescence in the dimethyl sulfoxide (DMSO)-treated controls remained consistently at low levels throughout the whole simulation period, although it was evident from the pSIVA^(TM)^-IANBD/PI ratio that especially co-cultured PCb GBM cells died at a slightly lower rate (Figure 1B,C,E,F). After stimulation with TMZ+AT101/AT101, both primary cultures showed a clear response to stimulation, as evidenced by a general increase in both green and red fluorescences in the cultures. Interestingly, after 3 days of stimulation, a predominance of red fluorescence was observed in the mono-cultures, while green fluorescence predominated in the co-culture indicating a predominant number of PS-positive cells or early apoptotic cells. After 6 days, another general increase in green and red fluorescences could be observed in both surviving mono- and co-cultured cells, with a predominance of green fluorescence under both conditions (Figure 1B,C,E,F and Appendix A). 

### 2.2. TMZ+AT101/AT101 Treatment Led to Phosphorylation of AKT, mTOR and GSK3ß, Resulting in Expression of Pro-Tumorigenic Genes in Surviving GBM Cells

To analyze the intracellular effects upon TMZ+AT101/AT101 therapy in surviving, MGMT promoter-methylated mono- and co-cultured GBM cells, respectively, we first performed a human MAPK (mitogen-activated protein kinase) phosphorylation antibody array of the respective GBM cells (PCa) after 3 days of stimulation (example depicted in Figure 2A). Here, GSK3β and mTOR, in particular, were present in high amounts in their phosphorylated forms in the GBM cells and continued to be induced upon therapy, especially in co-cultured GBM cells. Due to the selected antibodies used in the array, phosphorylation of mTOR reflected activation (phosphorylation at Ser2448), whereas that of GSK3β rather an inhibition (phosphorylation at Ser9) of the respective enzyme activities. In contrast to the alterations seen for mTOR and GSK3β, MAPK kinases and extracellular signal-regulated kinases (ERK) 1/2 appeared to play a minor role in the context of the TMZ+AT101/AT101 therapy.

To validate the results of the MAPK phosphorylation array, we performed Western blot analyses after stimulation of mono- and co-cultured GBM cells with TMZ+AT101/AT101 for 3 and 6 days, respectively (Figure 2B,C and Appendix A). 

Here, we focused on the analyses of phospho-Ser2448-mTOR and phospho-Ser9-GSK3β, and phospho-Akt, being involved in the regulation of mTOR and GSK3β and regulated by mTOR through phosphorylation of the Ser473 site [24,25,26]. 

PCa (Figure 2B) showed increased phosphorylation of Akt at Ser473 after 3 and 6 days of stimulation with TMZ+AT101/AT101 compared to controls, and this effect was equally detectable in both mono- and co-cultured GBM cells. For PCb (Figure 2C), this effect was more prominent in co-cultured GBM cells. Regarding the mTOR phosphorylation at Ser2448, a marked induction was observed in PCa after 6 days of the TMZ+AT101/AT101 treatment in mono- and co-cultured GBM cells, whereas in PCb this effect appeared already after 3 days, especially in co-cultured GBM cells. When looking at the phosphorylation of GSK3β at Ser9, PCa showed a clear induction in mono- and co-cultured GBM cells after 3 and 6 days, respectively, whereas in PCb this effect was mainly detectable after 6 days of stimulation in mono-cultured cells and after both time points examined in co-cultured GBM cells.

By promoting stemness, fostering EMT processes, or regulating angiogenesis or cell cycle properties [27,28,29,30,31,32], both GSK3β and mTOR are known regulators of GBM progression. Therefore, we next examined the expression of several pro-tumorigenic genes involved in these processes as a broader initial assessment to identify relevant processes involved in the development of chemoresistance towards treatment with TMZ+AT101/AT101 (Figure 3 and Appendix A). When analyzing the expression of the EMT markers β-catenin, L1 cell adhesion molecule (L1CAM), and snail family transcriptional repressor 2 (SNAIL2), the surviving PCa showed significant induction of all three markers upon stimulation with TMZ+AT101/AT101 after 3 or 6 days of treatment, respectively (Figure 3A). 

Depending on the gene investigated, the extent of induction varied between mono- and co-culture conditions but was preferentially observed under co-culture conditions. On the contrary, the surviving PCb showed no or only a slight regulation of the EMT markers analyzed after treatment with TMZ+AT101/AT101 in comparison to the controls (Figure 3B). With regard to stemness, we evaluated the expression of Nestin, octamer binding transcription factor 4 (OCT4), Tir nan Og (Nanog), Musashi homolog 1 (Musashi), and paired box protein 6 (Pax6), which were found to be upregulated upon treatment in both primary cultures to different extents depending on the culture conditions and the duration of the treatment (Figure 3A,B). The influence of the stimulation with TMZ+AT101/AT101 on the regulation of angiogenesis was evaluated by analyzing the expression of vascular endothelial growth factor a (VEGFa). PCa showed a significant increase in VEGFa expression after stimulation with TMZ+AT101/AT101 for 3 days in both mono- and co-cultures, which was less after 6 days of treatment. PCb on the other hand only showed an induction of VEGFa in mono-cultures after 3 days of stimulation. When looking at the expression cyclin-dependent kinase inhibitor 1A (CDKN1A, p21) and tumor protein p53 (p53) as important regulators of the cell cycle, we observed an increase in p21 expression levels in both PCa and PCb after 3 and 6 days of treatment with TMZ+AT101/AT101 in mono- and co-cultures, respectively. PCa showed a reduction of p53 levels in mono-cultures after 6 days of stimulation, whereas in co-cultures no regulation of p53 expression levels was found upon treatment with TMZ+AT101/AT101. PCb showed a reduced expression of p53 after 6 days of treatment in both mono- and co-cultures.

### 2.3. TMZ+AT101/AT101 Treatment of GSCs

In order to take the heterogeneous nature of GBM with the presence of various tumor cell subpopulations and their contribution to the development of chemoresistance into account, we analyzed the efficacy of TMZ+AT101/AT101 treatment in two patient-derived GSC primary cultures, GCSa and b, respectively (Figure 4 and Appendix A). Both GSCs were characterized by a methylated MGMT promoter (GSCa: 41.8%; GSCb: 72.8%). In accordance with our results regarding the differentiated GBM cells PCa and PCb, there was a prominent cytotoxic and antiproliferative effect of the treatment after 6 days yielding 74.14 ± 5.10% dead cells in GSCa and 75.52 ± 3.38% dead cells in GSCb, respectively (Figure 4A,C).

Next, we analyzed the regulation of stemness markers (Nestin, Oct4, Nanog, Musashi, Pax6) in surviving GSCa and GSCb cells in response to treatment with TMZ+AT101/AT101. In the case of GSCa, the stemness markers Nestin, Nanog, and Musashi were found to be slightly upregulated after 6 days of treatment with TMZ+AT101/AT101 (Figure 4B). Oct4 was upregulated after 3 days of treatment and the induction was further increased after 6 days of treatment. GSCb displayed an overall higher induction of the stemness markers compared to GSCa with upregulations of Nestin, Oct4, and Nanog observed after 3 days and more evidently after 6 days of treatment with TMZ+AT101/AT101 (Figure 4D).

### 2.4. Torin2-Mediated mTOR Inhibition Counteracts TMZ+AT101/AT101-Regulated Chemoresistance Mechanisms of Surviving GBM Cells

In order to prove the role of mTOR signaling and the resulting expression of pro-tumorigenic genes as a chemoresistance mechanism of surviving MGMT promoter-methylated GBM cells, co-cultured GBM cells (PCa) were treated with the mTOR inhibitor Torin2 in the presence or absence of TMZ+AT101/AT101 for 6 days (Figure 5). Torin2 is an ATP-competitive mTOR inhibitor, which inhibits the phosphorylation of both mTORC1 and mTORC2 [33]. Effective Torin2-mediated inhibition of mTOR phosphorylation was initially confirmed by Western blotting (Figure 5A). Treatment of co-cultured PCa with Torin2 for 6 days led to reduced p-mTOR levels in comparison to the controls, which was even more pronounced when treating the cells with a combination of Torin2 and TMZ+AT101/AT101 (Figure 5A).

In order to assess the Torin2-mediated mTOR inhibition and its influence on the mTOR pathway in more detail, we next analyzed the phosphorylation of the mTOR downstream substrates P70S6 kinase (P70S6K) and eukaryotic translation initiation factor 4E-binding protein 1 (4E-BP1). Treatment with TMZ+AT101/AT101 for 6 days led to increased p-P70S6K levels in comparison to the DMSO control. Treatment of PCa with the combination of Torin2 and TMZ+AT101/AT101 did not counteract the TMZ+AT101/AT101-mediated induction of p-P70S6K (Figure 5A). The phosphorylation of 4E-BP1 was not found to be regulated upon treatment with TMZ+AT101/AT101 for 6 days in comparison to the DMSO control. However, it seems that combined treatment with Torin2 and TMZ+AT101/AT101 led to slightly reduced p-4E-BP1 levels compared to the DMSO control (Figure 5A).

In order to evaluate, whether Torin2-mediated mTOR inhibition can increase the cytotoxic effect of TMZ+AT101/AT101 treatment, PCa was treated with TMZ+AT101/AT101, Torin2, and the combination of Torin2 and TMZ+AT101/AT101 for 3 and 6 days. Indeed, after 6 days, treatment with a combination of Torin2 and TMZ+AT101/AT101 yielded a 1.2-fold higher cell death rate (87.37 ± 2.15% dead cells) in comparison to treatment with TMZ+AT101/AT101 alone (74.73 ± 0.35% dead cells) (Figure 5B). Additionally, the combination of Torin2 and TMZ+AT101/AT101 led to a stronger decrease in the growth rate of PCa after 6 days of treatment compared to treatment with TMZ+AT101/AT101.

Next, we analyzed the regulation of genes involved in EMT (β-catenin, L1CAM, SNAIL2), stemness (Nestin, Oct4, Nanog, Musashi, Pax6), angiogenesis (VEGFa), and cell cycle regulation (p21, p53) in surviving PCa cells treated with TMZ+AT101/AT101, Torin2, or combined Torin2 and TMZ+AT101/AT101 (Figure 5C). The upregulation of gene expression observed for these genes upon treatment with TMZ+AT101/AT101 was found to be reduced when treating the cells with the combination of Torin2 and TMZ+AT101/AT101 after 3 or 6 days of treatment depending on the genes, respectively. Overall, this effect was more pronounced after 6 days of treatment.

### 2.5. TMZ+AT101/AT101 Treatment-Influenced Extracellular Vesicles Derived from Surviving GBM Cells

Given that GBMs are known to drive malignant progression through EVs exchange [11], and several studies have revealed the multi-dynamic role of EVs in the acquisition of drug resistance [34], we next examined whether combined TMZ+AT101/AT101 treatment has an impact on EVs derived from the primary GBM cells used in this study (Figure 6 and Appendix A). GBM cells were again cultured either as mono-culture or in co-culture with astrocytes or microglia. Here, after 3 days of stimulation, the cell culture inserts containing microglia and astrocytes were removed from the co-cultured GBM cells, the media was replaced by a growth medium supplemented with exosome-depleted fetal bovine serum, and the surviving GBM cells were incubated for additional 48 h to allow secretion of EVs before EV separation was performed by several processing steps (please refer to Section 4.8). 

First, we characterized the EVs of mono- and co-cultured GBM cells (PCa) with and without therapy, respectively, with respect to their size distribution and quantity by nanoparticle tracking analysis (NTA) (Figure 6A). Importantly, these investigations were performed on the whole EV population, i.e., before 0.2 µm filtration and isolation of CD63-positive EVS by magnetically activated cell sorting (MACS). We found that the number of EVs derived from mono- or co-cultured GBM cells increased upon TMZ+AT101/AT101 therapy (mono-culture: DMSO, 1.39–1.61 × 10^7^ particles/mL; TMZ+AT101/AT101, 5.05–5.69 × 10^7^ particles/mL; co-culture: DMSO, 1.11–1.22 × 10^7^ particles/mL; TMZ+AT101/AT101, 7.62–8.25 × 10^7^ particles/mL). This effect was particularly pronounced under co-culturing conditions. In addition, a shift in the mean EV size toward smaller EVs was observed upon therapy, an effect that was seen in both mono- and co-cultures (mono-culture: DMSO, mean 340 nm; TMZ+AT101/AT101, mean 330.5 nm; co-culture: DMSO, mean 297.5 nm; TMZ+AT101/AT101, mean 279 nm).

After performing a 0.2 µm filtration to concentrate especially small EVs and also a subsequent separation of CD63-positive EVs by MACS, we characterized the EVs with respect to the expression of the tetraspanin proteins CD9, CD63, and CD81, respectively, with and without chemotherapy. After verification of the successful 0.2 µm filtration by means of NTA (Figure 6B1, left), we showed that all three tetraspanins were basically detectable in the EVs, and as expected, CD63 signal was stronger in CD63-enriched (CD63+) EV fraction (Figure 6B1, right, data shown for EVs derived from mono-cultured PCa GBM cells). Interestingly, the CD63+ fraction also showed a strong CD81 signal and a clear CD9 signal. In addition, the CD63+ fraction demonstrated an increase in the CD63, CD81, and CD9 signals upon therapy, an effect that was only slightly or not at all pronounced in the CD63- flow-through fraction (Figure 6B1, right). Here, the tetraspanin signals showed the correct size in the Western blot (caveolin-1 as control), and the EVs were visible as small spherical structures in the negative staining transmission electron microscope after performing the 0.2 µm filtration (Figure 6B2; before CD63 MACS). 

Finally, we wondered whether some of the genes, which were previously studied and regulated upon the combination therapy with TMZ+AT101/AT101, were detectable in the separated CD63^+^ EVs. Indeed, the expression of β-catenin was detected in the EVs of mono- and co-cultured GBM cells. Moreover, the expression was increased specifically in EVs from treated, co-cultured GBM cells compared to the other conditions (Figure 6C). 

In summary, we showed that both mono- and co-cultured TMZ+AT101/AT101 surviving GBM cells were characterized by the pronounced appearance of PS-positive GBM cells over time, exhibited an activation of the PI3K/Akt/mTOR pathway as well as a regulation of the GSK3β with subsequent changes in the expression of pro-tumorigenic genes, and, moreover, were characterized by changes in the amount and composition of EVs.

## 3. Discussion

A pronounced chemoresistance, particularly in heterogeneous solid tumors, remains a major challenge in clinical practice. Singular therapeutic approaches often fail, e.g., due to the selection of resistant clones, which then leads to tumor recurrence. Therefore, multimodal therapeutic approaches are becoming increasingly important, ideally characterized by different mechanisms of action, and exerting the highest possible efficacy specifically on tumor cells [14,15,16]. Given this, in previous work, we investigated a combination therapy of TMZ and AT101 for the treatment of MGMT promoter methylated GBMs, which was also characterized by a sequential application and, thus, fewer side effects [17,18,19]. Although this approach demonstrated higher efficacy than, e.g., administration of TMZ alone, some GBM cells survived. Through the studies presented here, we have now shown that several aspects are relevant in the development of chemoresistance to the TMZ+AT101/AT101 therapy in surviving GBM cells.

First, using the Polarity Sensitive Indicator of Viability pSIVA^(TM)^ Kit, we found that PS-positive cells were predominant upon TMZ+AT101/AT101 therapy, particularly after 6 days of stimulation, although there was generally an increase in both PS- and PI-positive cells in the cultures compared with controls. Interestingly, this effect was observed in the co-cultures already after 3 days of stimulation, whereas it appeared in the mono-cultures only after 6 days of stimulation. 

PS is a negatively charged phospholipid predominantly located in the inner leaflet of the cell membrane in healthy cells. PS transfer between the inner and outer cell membranes is regulated by a group of ATPases, as well as by amino phospholipid translocases (flippases) [35]. PS exposure on the external leaflet of the plasma membrane is widely observed during apoptosis [35,36], including both healthy and tumor cells. Indeed, Zhao et al. [37] showed that increased levels of PS were detected in radiation-treated glioma xenografts using near-infrared optical imaging. In contrast to normal cells, cancer cells are limited in their ability to maintain PS asymmetry, resulting in PS being exposed at the surface of cell membranes [35]. Here, in addition to regulating cell death, PS is also involved in several other processes, such as immunomodulation. Moreover, PS could also be a driver of metastasis and a positive correlation between surface PC exposure of tumors and their malignancy is described [38]. Thus, when considering PS occurrence in TMZ+AT101/AT101-treated GBM cells over time, two aspects should be noted: First, the increased incidence of PS over time compared with PI suggests that more early than late apoptotic cells were present in the cultures. Since early apoptotic cells can in principle return to a physiological state, this is an indication that an increase in therapy-surviving GBM cells was observed during the cultivation period. On the other hand, a selection of GBM cells characterized by a particularly high PS content on the outside of the plasma membrane could in principle also occur upon therapy. However, both aspects would be an indication of a resistance mechanism to the TMZ+AT101/AT101 therapy.

Next, when looking at the intracellular effects of the TMZ+AT101/AT101 therapy, increased phosphorylation of especially Akt (Ser473), mTOR (Ser2448), and GSK3β (Ser9) was evident. These effects were observed in both surviving mono- and co-cultured GBM cells, respectively, although slight temporal differences in the detection of the corresponding phosphorylation signals were observed.

Indeed, antitumor drugs are well known to affect signaling pathways controlling apoptosis in GBM cells, thereby activating pro-survival mechanisms that render treatments ineffective [39]. For example, PI3K is a kinase that plays a central role in signaling pathways controlling cell survival, proliferation, motility, angiogenesis, and apoptosis [39]. Its activation phosphorylates the plasma membrane lipid phosphatidylinositol-4,5-biphosphate producing the second messenger phosphatidylinositol-3,4,5-triphosphate (PIP3). PIP3 induces the accumulation of signaling proteins such as Akt, which regulates downstream targets including GSK3β, Bad, and mTOR [39]. Notably, the PI3K/Akt signaling is deregulated in numerous GBMs [7,39]. 

mTOR itself is a serine/threonine protein kinase, which plays a key role in cell growth and homeostasis and is phosphorylated at Ser2448 via the PI3K/Akt signaling pathway [25]. mTOR acts as both a downstream effector and an upstream regulator, with signaling being carried out by two protein complexes known as mTORC1 and mTORC2. mTORC1 regulates cell growth through activation of P70S6K and inhibition of 4E-BP1, while mTORC2 phosphorylates Akt at Ser473 and then further takes part in cell survival, angiogenesis metabolism, proliferation, and cytoskeletal organization [33,39]. Since we were able to specifically observe the phosphorylation of Akt at position Ser473, as well as of mTOR at position Ser2448 upon the TMZ+AT101/AT101 therapy, it can be assumed that a regulatory loop between Akt and mTOR is of special interest in this context. The importance of mTOR signaling in the resistance mechanisms of surviving GBM cells towards TMZ+AT101/AT101 was underlined by our results regarding mTOR inhibition using Torin2. Torin2 is an ATP-competitive mTOR inhibitor of the quinoline class, which inhibits phosphorylation of both mTORC1 and mTORC2 and was shown to exhibit anti-tumor effects in several types of tumors [33]. Indeed, treatment of primary GBM cells with Torin2 reduced the phosphorylation of mTOR and this effect was even more pronounced when treating the cells with a combination of Torin2 and TMZ+AT101/AT101. In addition, treatment of primary GBM cells with TMZ+AT101/AT101 for 6 days only induced levels of p-P70S6K but the induction was not abolished by additional application of Torin2. On the contrary, treatment with TMZ+AT101/AT101 did not affect the phosphorylation of 4E-BP1 (Thr37/46). Indeed, the inhibitory effects of Torin2 on the phosphorylation of the mTORC1 substrate P70S6K have been shown to be concentration-dependent, with higher concentrations needed for the abolishment of the phosphorylation of P70S6K [33,40]. However, due to the high sensitivity of the astrocytes used for co-cultures towards Torin2 (~30% cell death after 6 days of treatment), we were restricted to using low concentrations of Torin2. Despite this, we observed a sufficient inhibition of mTOR phosphorylation when treating primary GBM cells with 0.5 nM Torin2, and treatment with Torin2 together with TMZ+AT101/AT101 yielded an increased cytotoxic effect in comparison to treatment with solely TMZ+AT101/AT101. 

GSK3β is a ubiquitously expressed serine/threonine protein kinase that functions as a key regulator of the canonical Wnt/β-catenin pathway [41] and also as a critical downstream element of the PI3K/Akt cell survival pathway. Here, GSK3β controls the intracellular localization and degradation of cyclin D1 by phosphorylation at threonine-286 and of β-catenin by phosphorylation at threonine-41, serine-33 and -37, resulting in its ubiquitin-dependent degradation at the proteasome. Importantly, the GSK3β activity can be inhibited by Akt-mediated phosphorylation at Ser9 leading to the translocation of β-catenin to the nucleus [24], which in turn promotes several pro-tumor processes, including EMT, proliferation, migration, and invasion, as well as the maintenance of a stem cell-like cell fraction [41]. Since we observed increased phosphorylation of GSK3β at position Ser9, this, as well as the phosphorylation of Akt and mTOR already discussed, suggests an escape mechanism of the treated GBM cells, which ultimately promotes tumor cell survival even upon TMZ+AT101/AT101 therapy. This effect can also be underlined by the fact that both kinases are known to be involved in partially connected intracellular regulatory mechanisms and are also able to regulate each other [42]. 

In order to obtain a general overview of the processes involved in the development of chemoresistance towards treatment with TMZ+AT101/AT101 we examined the expression of several pro-tumorigenic genes involved in processes such as EMT, stemness, angiogenesis, and cell cycle regulation initially in an exploratory manner. Consistent with our results so far, we observed the regulation of several pro-tumorigenic genes upon the TMZ+AT101/AT101 therapy. Specifically, genes involved in EMT [43], such as β-catenin, L1CAM, fibronectin, and SNAIL2, as well as genes involved in stemness [44], such as Nestin, Oct4, Musashi, Pax6, and Nanog, were significantly induced upon the TMZ+AT101/AT101 therapy in surviving GBM cells. Depending on the GBM cells investigated (PCa or PCb), stronger regulation was observed in mono- or co-cultures after 3 or even 6 days, respectively, and not all genes were regulated in both primary cultures, and if they were, not always to the same intensity.

Both GSK3β and mTOR are known regulators of stem cell and EMT processes leading to GBM progression [27,28,29,30,31,32,42]. For example, Garros-Regulez et al. [29] showed that a genetic and pharmacological mTOR inhibition decreased the expression of SOX2 and SOX9 in GBMs, which both are known to regulate stem cell properties. Friedmann et al. [28] demonstrated that mTOR is involved in the maintenance of GSCs. For GSK3β, Korur et al. [30] presented that inhibition of the GSK3β activity induced GBM cell differentiation. Further, the formation of neurospheres was impaired, and clonogenicity was reduced in a dose-dependent manner. In addition, Joshi et al. [45] showed that FoxM1, which promotes β-catenin nuclear translocation by directly binding to β-catenin, promoted self-renewal and chemoresistance of GBM cells [31,45]. Regarding the regulation of EMT processes, the long noncoding RNA LINC-PINT is known to inhibit the EMT of GBMs by interfering with the Wnt/β-catenin signaling [46], whereas the microRNA miR205 targets ZEB1 and, thus, affecting EMT [47] via the Akt/mTOR signaling pathway. Here, ZEB1 is a transcription factor that mediates EMT in a variety of tumors, including gliomas [48]. 

In addition to the mentioned EMT and stemness genes, VEGFa, a well-known gene responsible for angiogenesis [49], and different cell cycle regulators such as p21 and p53 [50] were regulated in TMZ+AT101/AT101-treated surviving GBM cells. As stated before, the regulation depended on the primary culture investigated, the culture conditions, and the time points analyzed. 

Indeed, the PI3K/Akt/mTOR signaling has been identified as a key contributor to GBM angiogenesis [27]. For example, the inhibition of HIF1α (hypoxia-inducible transcription factor 1α) and mTOR signaling pathways mediated by rapamycin or mTOR silencing has been shown to reduce the ability of GBM cells to acquire genetic and/or phenotypic characteristics of endothelial cells, termed vasculogenic mimicry or transdifferentiation [27,51]. Moreover, the frequent loss of PTEN (phosphatase and tensin homolog) in GBMs, which physiologically inhibits cell growth and can downregulate the PI3K/Akt/mTOR pathway, led to the expression of VEGF receptor 2 in GBMs and, thus, contributed to the failure of anti-angiogenesis treatments in these tumors [52]. Interestingly, Akt is also able to upregulate HIF1α indirectly via the inhibition of the GSK3β activity due to the Ser9 phosphorylation [53]. HIF1α is a potent activator of angiogenesis and invasion by upregulating several target genes, including VEGFa.

When looking at the expression of different cell cycle regulators upon TMZ+AT101/AT101 therapy, the strong induction of p21 expression was particularly striking. Here, p21 is physiologically known as a specific inhibitor of the cyclin-dependent kinases, which in turn are well known to drive the cell through the cell cycle and thereby promote cell growth [50]. Indeed, under stress conditions, p21 expression is increased through p53-dependent and -independent pathways [54]. P21 levels may increase in a p53-dependent manner and as a result, contribute to arresting cell proliferation in response to DNA damage and many other cellular stressors. However, all GBM cultures used in this study are characterized by p53 missense mutations in the DNA binding region leading to a non-functional p53 protein and the inhibition of transcription factor activity (PCa: p.R273H; PCb: p.R248Q) [55]. In these tumor cells, the cell cycle checkpoint function of p21 is lost due to p53 mutations. Thus, p21 might then be phosphorylated and stabilized by Akt, resulting in cell survival by p21-mediated inhibition of apoptosis [56]. P21 regulates multiple cellular processes, including apoptosis, differentiation, and quiescence [57]. Since the main function of mTOR is to promote cell growth [58], an enhanced PI3K/Akt/mTOR activation concomitant with high expression of p21 initially appears contradictory. However, it was reported that p21 levels were affected by mTORC1 through an increased mRNA translation [59], as well as by a specific increased translation of p21 mRNAs [60,61,62]. Moreover, it was shown that upon mTORC1 activation, the 4E-BP1/p21 complex was disrupted and p21 was stabilized resulting in a mTORC1-mediated induction of p21 [61]. Thus, the observed regulation of p21 expression, such as factors involved in EMT, stemness, or angiogenesis, appear to be linked to the resistance mechanisms of surviving GBM cells induced upon TMZ+AT101/AT101 treatment. Inhibition of the mTOR phosphorylation by treatment with the mTOR inhibitor Torin2 combined with TMZ+AT101/AT101 counteracted the TMZ+AT101/AT101-mediated induction of the pro-tumorigenic genes analyzed, highlighting the involvement of mTOR signaling in the development of chemoresistance mechanisms.

Since the tumor heterogeneity and the presence of various tumor cell subpopulations are known contributors to the development of chemoresistance, we next examined the therapeutic efficacy of sequentially applied TMZ+AT101/AT101 on two patient-derived primary GSC cultures. Fortunately, treatment with TMZ+AT101/AT101 yielded strong cytotoxic effects in GSCs after 6 days of treatment, indicating the suitability of this treatment regimen to effectively target different tumor cell subpopulations. In line with the results previously published by our group, stem-like cells exhibit higher sensitivity towards treatment with AT101, whereas a sole treatment with TMZ yields low cytotoxic effects (GSCa: 12.74 ± 1.02% dead cells upon TMZ and 72.39 ± 1.70% dead cells upon AT101 treatment for 6 days; GSCb: 49.61 ± 2.98% dead cells upon TMZ and 63.93 ± 4.06% dead cells upon AT101 treatment for 6 days) [63]. On the contrary, primary GBM cells were found to exhibit a higher sensitivity towards treatment with TMZ (PCa: 74.46 ± 12.28% and PCb: 72.04 ± 1.94% dead cells after 6 days of treatment) compared to treatment with AT101 (PCa: 7.38 ± 7.57% and PCb: 17.67 ± 2.81% dead cells after 6 days of treatment). Even though the sequential treatment with TMZ+AT101/AT101 yielded high cytotoxicity in both more differentiated GBM cells and GSCs, some GBM cells manage to evade the treatment regimen and survive by developing various chemoresistance mechanisms. Based on our results presented in this study, the upregulation of various stemness markers appears to be one possible chemoresistance mechanism allowing the GBM cells to survive. The upregulation of the stemness markers Nestin, Oct4, Nanog, Musashi, and Pax6 was observed in both primary GBM cells and GSC cultures, with varying extend and dependent on the time point of treatment. Even though GSCs display an overall higher basic expression of stemness markers in comparison to more differentiated GBM cells, treatment with TMZ+AT101/AT101 for 6 days still further induced the expression levels of the markers investigated. Thus, our results underline the role of stemness regulation on the development of chemoresistance mechanisms in GBM cells, which includes the presence of a therapy-resistant subpopulation of GSCs or the induction of dedifferentiation in the non-GSC subpopulation of GBM cells as a response to the treatment.

Another important factor determining the therapeutic efficacy of treatment regimens containing TMZ is the MGMT promoter methylation status. TMZ as an alkylating agent damages DNA by adding methyl groups to the N7 and O6 positions of guanine and the N3 positions of adenine [64,65,66]. This methylation leads to an inaccurate pairing of the methylated guanine with thymine during replication, resulting in DNA double-strand breaks, irreparable genomic damage, and cell death [67]. The MGMT activity reverses the alkylation-induced DNA damage, thus, diminishing the cytotoxic effect of TMZ [67]. Therefore, MGMT promoter methylation is linked to patient outcomes with higher MGMT promoter methylation resulting in better response to TMZ treatment [68,69]. All GBM cultures investigated in this study exhibit different extends of MGMT promoter methylation (PCa: 91.8%; PCb: 84.2%; GSCa: 41.8%; GSCb: 72.8%) and are therefore in principle able to respond to a TMZ treatment. In the case of GBM cells with unmethylated MGMT promoters, the chemoresistance mechanisms in general might differ compared to GBM cells with methylated MGMT promoters [70]. Whether the sequentially applied combination of TMZ+AT101/AT101 is efficient in GBM cells with unmethylated MGMT promoters and which chemoresistance mechanisms might be developed by GBM cells upon the therapy can be the focus of follow-up studies. In addition, possible resistance mechanisms of GBM cells upon combined treatment with TMZ+AT101/AT101 and radiotherapy might differ from the chemoresistance mechanisms described in the present study, since it is known that GBM cells are characterized by specific resistance mechanisms towards radiotherapy [71]. Thus, a detailed analysis of these resistance mechanisms is beyond the scope of this manuscript and has to be evaluated in follow-up studies.

Since several studies have shown a multi-dynamic role of EVs in the acquisition of drug resistance [34,72], we next separated and characterized EVs derived from mono- or co-cultured GBM cells, respectively, upon TMZ+AT101 therapy. 

EVs are a heterogeneous population of lipid-bilayer delimited vesicles secreted by cells into the extracellular space. Various subpopulations of EVs such as microvesicles, exosomes, and apoptotic bodies are known and are differentiated based on their biogenesis, release pathways, size, content, and biological function [73]. However, in line with the recommendation stated by the International Society for Extracellular Vesicles in 2018 (Misev2018), EVs can be named according to their size, as small EVs (diameter < 200 nm) and large EVs (diameter > 200 nm) [74,75]. In detail, exosomes are formed by the inward budding of the limiting membrane of early endosomes and are typically ~ 30 nm up to 150 nm in diameter [73]. The size of microvesicles ranges from ~50 nm up to 1.5 µm in diameter and they are formed by direct outward budding of the cell’s plasma membrane, whereas apoptotic bodies are released by dying cells into the extracellular space and range in size from ~50 nm up to 2 µm in diameter [73]. 

In GBMs, Panzarini et al. [74] showed that the number of EVs and the ratio between small and large EVs changed depending on the GBM cell lines investigated, with large EVs being more numerous than small EVs regardless of the GBM cells studied. Here, small EVs had an average diameter of 85 ± 1.2 nm and a range diameter of 35–150 nm. The size of large EVs was more heterogeneous, with an average between 282 ± 70 nm to 430 ± 168 nm [74]. In our study, the mean EV size in the total EV fraction was 340 nm for mono-cultured and 297 nm for co-cultured, untreated GBM cells (DMSO controls), respectively. Thus, the mean EV size was within the range of known data, whereby the number of large EVs appeared to predominate. After performing a 0.2 µM filtration, the mean size of EVs was approximately 142 nm (example for co-cultured GBM cells), thus falling within the range of small EVs as expected. Interestingly, a shift in the mean EV size in the total EV fraction toward smaller EVs was observed upon therapy in both surviving mono- and co-cultured GBM cells (mono-culture TMZ+AT101: 330 nm; co-culture TMZ+AT101: 279 nm), respectively, and the number of EVs derived from mono- or co-cultured GBM cells increased upon TMZ+AT101 therapy. 

After performing several processing steps including 0.2 µm filtration, CD63 MACS, and Western/dot blotting, we found CD63, CD81, and CD9 signals in the CD63^+^ fraction, indicating the existence of probably CD63^+^/CD81^+^ and CD63^+^/CD9^+^ double-positive EVs. In addition, the amount of tetraspanin proteins increased upon therapy. Here, we used caveolin-1 as an internal control for quantification in the Western/dot blots. Indeed caveolin-1 is known to be the major component of the so-called caveolae, which as a characteristic feature of the plasma membrane can internalize and fuse with early endosomes, and thus are also found in EVs including exosomes [76,77]. 

Kıyga et al. [78] showed that the amount of CD63 in the EVs of TMZ (200 µM)-treated GBM cells increased, however, without any statistically significant difference. Ricklefs et al. [79] demonstrated that EVs with double positive tetraspanin expression (CD63^+^/CD81^+^; CD9^+^/CD81^+^) were enriched in GBM cell lines. Moreover, the total numbers of double-positive CD63^+^/CD81^+^ and CD63^+^/CD9^+^ EVs were increased in GBM patients, with the combination of CD63^+^/CD81^+^ being the most significant [79]. Here, the tetraspanins CD9, CD63, and CD81 are considered to be specific EV markers that are ubiquitously present on EVs from most cell types [79,80]. They are commonly found in exosomes, however, are not specific to these vesicles [81]. 

Furthermore, in this study, we were able to show that β-catenin was found at high levels in the filtrated CD63^+^ fraction, with mono-cultured GBM cells with/without therapy showing the same amount of β-catenin in the EVs, whereas a higher amount of β-catenin was detectable in the EVs of co-cultured GBM cells upon therapy compared to the unstimulated controls. 

In fact, due to their ability to protect and transfer biological cargo consisting of proteins, lipids, and nucleic acids to recipient cells, EVs play a crucial role in cellular communication and tumorigenesis [82]. Indeed, several studies uncovered the role of EVs in the development and progression of GBMs. For example, Ricklefs et al. [83] showed that EVs from GBMs are involved in the maintenance of the intratumoral heterogeneity, and Panzarini et al. [74] demonstrated that EVs derived from GBMs modulated the activation of macrophages toward a tendentially M2-like phenotype. Further, Ricklefs et al. [84] presented that the programmed cell death 1 ligand was present in EVs, which could systemically suppress an anti-tumor immunity by inhibiting T cell activation. In addition, he reported that the fatty acid synthase can be detected in EVs derived from GBMs [85]. Moreover, exosomes derived from GBM cells have been shown to be enriched in oncogenic proteins (EGFRvIII) [86] and angiogenic factors [87]. Interestingly, André-Grégoire et al. [88] demonstrated that TMZ treatment led to the enrichment of EVs with cargoes dedicated to cell adhesion processes, and Lombardi et al. [89] documented that the cyclooxygenase-2, which was upregulated by TMZ, was transported by EVs. In addition, Ma et al. [90] showed that gossypol was a potent activator of EV secretion, and Pan et al. [91] presented data showing that EVs derived from GBMs promoted the proliferation and migration of neural progenitor cells via the PI3K-Akt pathway. Now, we are able to show that a higher amount of β-catenin was detectable in the EVs of co-cultured GBM cells upon the TMZ+AT101 treatment. Since, as mentioned above, the expression of EMT markers such as β-catenin is regulated via the GSK3β and mTOR signaling pathways [27,28,29,30,31,32,42], this observation is consistent with our previous results. 

Overall, our investigations showed that even when chemotherapeutic agents of different effect specificity were used, a variety of chemoresistance mechanisms of surviving GBM cells must be assumed (for a summary of the identified mechanisms see Figure 7), an aspect that can only be circumvented by further sophisticated multimodal (local) therapeutic approaches.

## 4. Materials and Methods

### 4.1. Cultivation of Primary GBM Cells, Patient-Derived Glioma Stem-like Cells (GSCs), and Healthy Brain Cell Lines

Cultured human primary GBM cells and patient-derived glioma stem-like cells (GSCs) were generated by dissociation of tumor material obtained by surgical dissection at the Department of Neurosurgery (Kiel, Germany) with approval of the ethics committee of the University of Kiel, Germany, after written informed consent of donors (file reference: D471/15 and D524/17) and in accordance with the Helsinki Declaration of 1975. The obtained human primary GBM cells were cultured as previously described [19]. GSCs were cultured under stem-like cell conditions in F12 media supplemented with B27 supplement (Thermo Fisher Scientific, Waltham, MA, USA), 2 mM L-glutamine, and 1% penicillin–streptomycin (10,000 U/mL). The growth factors EGF and bFGF were added at a concentration of 10 ng/mL. GSCs were characterized by the formation of neurospheres, the ability to survive and proliferate under stem cell conditions, and to differentiate into more mature cells, which was proven as described before [92,93,94]. The purity of the GSCs was ascertained by immunostaining with cell type-specific markers and by the absence of contamination with mycoplasms. 

The human microglial cell line HMC3 (ATCC CRL-3304) was purchased from the American Type Culture Collection (ATCC, Manassas, VA, USA). The human fetal astrocyte cell line SVGA was kindly provided by the group of Christine Hanssen Rinaldo, University Hospital of North Norway [95] with the permission of W.J. Altwood [96]. All primary cells or cell lines were cultured in Dulbecco´s modified Eagle´s medium (DMEM; Life Technologies, Carlsbad, CA, USA) supplemented with 10% fetal bovine serum (FBS; PAN-Biotech GmbH, Aidenbach, Germany), 1% penicillin–streptomycin (10,000 U/mL; Thermo Fisher Scientific), and 2 mM additional L-glutamine (Thermo Fisher Scientific). Cells were routinely checked for mycoplasma contamination, and for identity by immunocytochemistry staining of cell-type specific markers. 

### 4.2. Stimulation of Cells

For stimulation of mono- and co-cultured GBM cells, primary tumor cells were seeded in 10 cm culture dishes with 150,000 cells/dish in DMEM without phenol-red (PAN-Biotech) supplemented with 10% FBS, 1% penicillin–streptomycin and 2 mM L-glutamine (from now on referred to as complete medium) and allowed to adhere overnight at 37°/5% CO_2_. Co-cultures mimicking the incomplete GBM resection were prepared as described before [19]. In brief, SVGA (2600 cells) and HMC3 (60,000 cells) were seeded on coverslips (Ø 18 mm) in Millicell^®^ cell culture inserts (0.4 µm, Merck Millipore, Burlington, MA, USA) and were allowed to adhere in 30 µL medium for three hours before the flooding of the cell culture inserts with 2 mL medium. On the following day, cells were washed with phosphate-buffered saline (PBS), and the medium was exchanged to complete medium containing 50 µM temozolomide (TMZ; stock dissolved at 100 mM in dimethyl sulfoxide (DMSO; Sigma-Aldrich, St. Louis, MO, USA); Merck Millipore) and 2.5 µM AT101 (stock dissolved at 100 mM in DMSO; Tocris, Bristol, UK), and cultured for three days. On day three, the cells were washed with PBS, the medium was exchanged to complete medium containing solely 2.5 µM AT101, and the cells were cultivated for another three days. GSCs were stimulated under the same conditions and for the same periods, but under stem-like cell conditions in F12 media supplemented with B27 supplement (Thermo Fisher Scientific), 2 mM L-glutamine, and 1% penicillin–streptomycin (10,000 U/mL). The growth factors EGF and bFGF were added at a concentration of 10 ng/mL. Controls were stimulated with equal volumes of DMSO, and for inhibition of mTOR 0.5 nM Torin2 (stock dissolved at 10 mM in DMSO; Selleck Chemicals GmbH, Planegg, Germany) was added in the presence or absence of TMZ and AT101 in additional samples for up to six days. Cytotoxicity assay, Western blotting, and analysis of gene regulation (see below) were performed after three and six days of stimulation, respectively.

### 4.3. Cytotoxicity Assay

The cytotoxic effects were determined using the CytoTox-Fluor^TM^ Cytotoxicity Assay (Promega, Madison, WI, USA) according to the manufacturer’s instruction and as described before [19,63]. Briefly, supernatants of treated and control cells were collected at days three and six of stimulation, mixed with the bis-AAF-R110 substrate, and measured in a fluorescence microplate reader (Infinite M200Pro, TECAN, Zürich, Switzerland) at 485/535 nm. The exact numbers of dead cells were determined according to a prepared standard of digitonin-lysed (82.5 µg/mL; Merck Millipore) cell dilutions of each cell line, respectively. Moreover, the cell survival was determined by counting of viable cells with a hemocytometer at days zero, three, and six of the treatment. The percentages [%] of dead cells were calculated as the n-fold number of viable cells as described in Equations (1) and (2) after three and six days of stimulation, respectively. Growth rates were calculated as an n-fold number of alive cells compared to day zero of the treatment.
(1)$$\mathrm{Dead}\;\mathrm{cells}\;(\mathrm{day}\;3)\;[\%]=\frac{number\;of\;dead\;cells\;\left[day\;3\right]}{number\;of\;dead\;cells\;\left[day\;3\right]+vital\;cells\;\left[day\;3\right]}\times 100$$
(2)$$\mathrm{Dead}\;\mathrm{cells}\;(\mathrm{day}\;6)\;[\%]=\frac{number\;of\;dead\;cells\;\left[day\;3+day\;6\right]}{number\;of\;dead\;cells\;\left[day\;3+day\;6\right]+vital\;cells\;\left[day\;6\right]}\times 100$$

### 4.4. pSIVA Real-Time Apoptosis Microscopy

The Polarity Sensitive Indicator of Viability Apoptosis Detection Kit (pSIVA-IANBD, Bio-Techne GmbH, Minneapolis, MN, USA) was used for quantification of apoptosis of primary GBM cells undergoing stimulation with 50 µM TMZ and 2.5 µM AT101 for three or six days, as described above. For the purpose of live cell imaging and the restrictions given by the setup of the microscope, the mono- and co-cultures of primary GBM cells had to be scaled down to 6 well-plates regarding the total number of cells used while maintaining the ratio of primary GBM cells and healthy brain cells (SVGA and HMC3). Therefore, primary GBM cells were seeded at 25,000 cells/well. For co-cultures, HMC3 (10,500 cells) and SVGA (400 cells) were seeded on coverslips (Ø 10 mm) placed in Millicell^®^ cell culture inserts (0.4 µm, Merck Millipore) and were allowed to adhere in 10 µL medium for three hours before flooding of the cell culture inserts with 1.5 mL medium as described above. The pSIVA-IANBD Kit was used according to the manufacturer’s instructions. In brief, 10 µL/mL of pSIVA™-IANBD and 5 µL/mL of propidium iodide (PI) staining solutions were added to the cultures 24 h after stimulation. After incubation of the cells with the dyes for 15 min, the fluorescence was imaged using the Lionheart FX automated microscope (BioTek, Bad Friedrichshall, Germany) using the green fluorescence filter set for pSIVA™-IANBD (EX 488 nm/EM 530 nm) and a red fluorescence filter set for PI (EX 520 nm/EM 585 nm). Additionally, bright field images were taken to assess the total number of cells within the respective frames. Images were taken every 4 h with 10-fold magnification over the time course of the stimulation for up to 6 days. The Gen5 Data analysis software (BioTek) was used to quantify the total numbers of pSIVA™-IANBD and PI-positive cells in relation to the total number of cells within the frame.

### 4.5. Quantitative Reverse Transcription-Polymerase Chain Reaction (qRT-PCR)

RNA of cells was isolated using the TRIzol^®^ Reagent (Thermo Fisher Scientific) or with the ARCTURUS^®^ PicroPure^®^ RNA Isolation Kit (Applied Biosystems, Waltham, MA, USA) according to the manufacturer´s instructions. DNA digestion, cDNA synthesis, and qRT-PCR were performed as described before [17,19,63] using TaqMan primer probes (Applied Biosystems): β-catenin (Hs00172016_m1), glycerinaldehyde 3-phosphate dehydrogenase (GAPDH) (Hs99999905_m1), L1 cell adhesion molecule (L1CAM) (Hs00240928_m1), musashi homolog 1 (Musashi) (HS00159291_m1), Tir nan Og (Nanog) (Hs02387400_g1), Nestin (Hs00707120_s1), octamer binding transcription factor 4 (OCT4) (Hs00999632_g1), paired box protein 6 (Pax6) (Hs01088114_m1), cyclin-dependent kinase inhibitor 1A (CDKN1A, p21) (Hs00355782_m1), tumor protein 53 (p53) (Hs00153340_m1), snail family transcriptional repressor 2 (SNAIL2) (Hs00950344_m1), vascular endothelial growth factor a (VEGFa) (Hs00173626_m1). Cycles of threshold (C_T_) were determined and ∆C_T_ values of each sample were calculated as C_T_gene of interest—C_T_GAPDH. The induction of gene expression upon stimulation is displayed as n-fold expression changes 2^∆C^_T_
^control−∆C^_T_
^stimulus^.

### 4.6. Mitogen-Activated Protein-Kinase (MAPK) Antibody Array

To compare the changes in activation/phosphorylation of MAPK between stimulated primary GBM cells and the respective controls, the Human MAPK Phosphorylation Antibody Array (ab211061, Abcam, Cambridge, MA, USA) was used according to the manufacturer´s protocol. The array consists of a nitrocellulose membrane containing 17 anti-MAPK antibodies spotted duplicates, including positive and negative controls as well as a blank. Cell lysates were prepared from differentially stimulated primary GBM cells as described before [17,63]. After incubation of the membranes with 2 mL of blocking buffer for 30 min at room temperature (RT), 200 µg of total protein per sample (1 mL total volume, in blocking buffer) were added onto the membranes and incubated overnight at 4 °C. The antibody array membranes were washed, subsequently incubated with the Detection Antibody Cocktail, and incubated for 1.5 h at RT. The membranes were washed followed by incubation with HRP-Anti-Rabbit IgG for 2 h at RT. After washing, the membranes were subjected to visualization with a chemiluminescence-based detection method. The signal densities were quantified using ImageJ^®^ software (Version 1.54a). The signals were normalized to the positive control provided on the antibody array membranes. 

### 4.7. Western Blot

Differentially treated cells were harvested, and 3 to 10 µg of protein per sample was used for Western blotting as described before [63,97]. Used primary antibodies were anti-phospho-GSK3b (1:250, rabbit; #9336, Cell Signaling, Danvers, MA, USA), anti-phospho-Akt (1:250, rabbit; #4060, Cell Signaling), anti-phospho-mTOR (1:250, rabbit; #2971, Cell Signaling), anti-phospho-4E-BP1 (1:1000, rabbit; #2855, Cell Signaling), and anti-phospho-P70S6K (1:1000, rabbit; #9205, Cell Signaling) in 5% bovine serum albumin/tris-buffered saline with 0.1% Tween 20 (TBST). Secondary antibody was donkey-anti-rabbit IgG-HRP (1:12,500; A16035, Thermo Fisher Scientific) in 2% (*w*/*v*) casein/TBST. Loading of equal amounts of protein was confirmed by stripping and incubation of the membranes with anti-GAPDH (1:200, mouse; sc-47724, Santa Cruz Biotechnology, Dallas, TX, USA) in 2% (*w*/*v*) casein/TBST and the secondary antibody donkey-anti-mouse IgG-HRP (1:10,000; A16011, Thermo Fisher Scientific) in 2% (*w*/*v*) casein/TBST. The signal densities were quantified using ImageJ^®^ software. The signals were normalized to GAPDH, and the n-fold signal induction was determined in relation to the respective control samples (control = 1).

### 4.8. Separation and Characterization of Extracellular Vesicles (EVs)

The separation and characterization procedure of the EVs with all steps is shown in Figure 8. In detail, EVs were separated from the cell culture supernatant of primary GBM cells stimulated with 50 µM TMZ + 2.5 µM AT101 for three days using the Total Exosome Isolation Reagent (Thermo Fisher Scientific) according to the manufacturer´s protocol. After three days of stimulation, the cells were washed with PBS, the cell culture inserts containing HMC3 and SVGA were discarded in the setting of co-cultured GBM cells, and the medium was replaced by DMEM without phenol-red supplemented with 10% exosome-depleted FBS (Thermo Fisher Scientific). The cells were incubated for 48 h to allow for the secretion of EVs (cell density 50–70%). Subsequently, 10 mL of the media were harvested and cleared by two consecutive centrifugation steps at 300× *g* for 10 min and 2000× *g* for 30 min at 4 °C to remove cells and debris. Cleared supernatant was transferred to a new tube and 5 mL of the Total Exosome Isolation reagent was added and mixed well by vortexing until a homogenous solution was obtained. The samples were incubated overnight at 4 °C. On the following day, the samples were centrifuged at 10,000× *g* for 1 h at 4 °C. The supernatant was discarded, and the pellet was resuspended in 1 mL of 0.1 µm filtered PBS. Particle concentration and size distribution of separated EVs were determined by nanoparticle tracking analysis (NTA, Nanosight NS300, Malvern Instruments, Malvern, UK). 

Further purification of the obtained EVs and enrichment for small vesicles was achieved by 0.2 µm filtration followed by using the human Exosome Isolation Kit CD63 (Miltenyi Biotec, Bergisch Gladbach, Germany). In order to confirm the enrichment for small vesicles after filtration and to characterize the obtained vesicular fraction with regard to the particle concentration and size distribution, the obtained EVs were subjected to NTA prior to further purification steps. The enrichment was performed by positive selection using MicroBeads recognizing the tetraspanin protein CD63 and was completed according to the manufacturer´s protocol. Briefly, EVs were magnetically labeled by incubation with CD63 magnetic beads for 1 h at RT. Subsequently, the labeled EVs are loaded onto a column, which is placed in the magnetic field of a µMACS™ Separator. The magnetically labeled EVs are retained within the column, while the unlabeled vesicles and cell components run through the column. After removing the column from the magnetic field, the intact EVs were either collected by elution with isolation buffer (analysis by RT-qPCR) or directly lysed with lysis buffer [93,94] in the column for Western blotting analysis. For both downstream analyses, the EVs isolated from *n* = 5 biological replicates were pooled.

The CD63 enriched fraction of EVs obtained after magnetically activated cell sorting (MACS) separation was characterized regarding the expression profile of the tetraspanin proteins CD9, CD63, and CD81 by means of Western blotting. Used primary antibodies were anti-CD9 (1:250, mouse; ab2215, Abcam), anti-CD63 (1:500, mouse; ab59479, Abcam), anti-CD81 (1:250, mouse; sc-23962, Santa Cruz Biotechnology), and anti-caveolin-1 (1:200, rabbit; sc-894, Santa Cruz Biotechnology) in 2% casein/TBST, secondary antibodies were donkey-anti-rabbit IgG-HRP (1:12,500; A16035, Thermo Fisher Scientific) and donkey-anti-mouse IgG-HRP (1:10,000; A16011, Thermo Fisher Scientific) in 2% (*w*/*v*) casein/TBST. The signal densities were quantified using ImageJ^®^ software. The signals were normalized to caveolin-1 and the n-fold signal induction was determined in relation to the respective control samples (control = 1).

The morphology of separated EVs was evaluated exemplarily by negative staining transmission electron microscopy (TEM). The negative staining samples were prepared by using glow-discharged carbon-coated electron microscopy (EM) grids (Electron Microscopy Sciences, Hatfield, PA, USA). The glow discharge was performed by using the Mini Sputter Coater System (Quorum Technologies, Lewes, UK), with a strong current of 25 mA for 30 s. The EVs were separated from cell culture supernatant as described above and the obtained vesicle pellet was resuspended in 100 mM Tris–HCl (pH 6.8). EV samples were added to the glow-discharged carbon-coated EM grid for 30 s, and the access volume was removed using filter paper. The EM grid was stained twice with 1% aqueous uranyl acetate solution (Merck Millipore), blotted again with filter paper, and air dried. The micrographs were collected using the JEOL-1400 Plus TEM (JEOL, Tokyo, Japan) operating at 100 kV with a 50,000-fold magnification with the TemCam-F416 Camera (TVIPS GmbH, Gauting, Germany).

### 4.9. Statistical Analysis

The data were statistically analyzed using the GraphPad Prism 8.4^®^ software (GraphPad Software, San Diego, CA, USA). Depending on the experimental setup either a Student *t*-test, a one-way analysis of variance (ANOVA), or a two-way ANOVA was performed, as indicated for each experiment in the respective figure legends. The sample size and the number of replicates are stated in the figure legends. The data are generally presented as mean ± standard deviation (SD). Statistical significance is marked with asterisks depending on the *p*-value: * *p* < 0.05, ** *p* < 0.01, and *** *p* < 0.001.

## Figures and Tables

**Figure 1 ijms-24-09075-f001:**
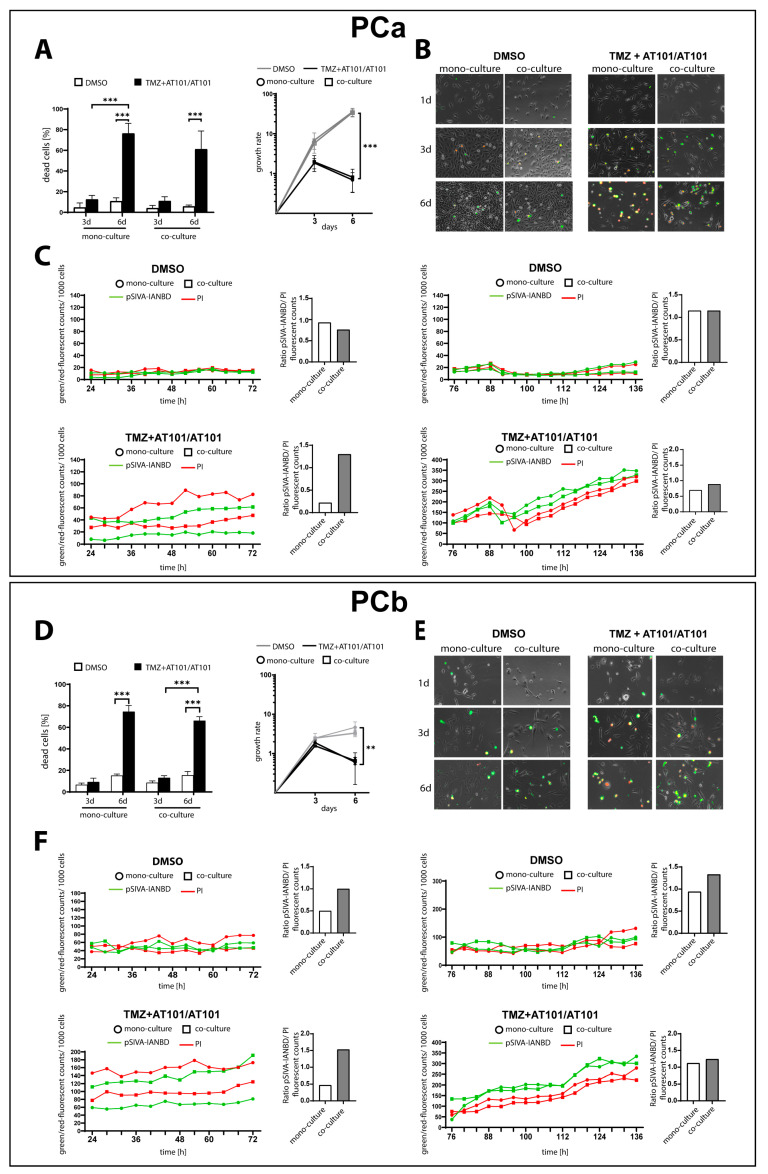
TMZ+AT101/AT101 treatment led to a predominance of phosphatidylserine (PS)-positive GBM cells over time. Human primary GBM cells were mono-cultured or co-cultured with microglia and astrocytes in defined cellular proportions mimicking an incomplete GBM resection. The cultures were treated with TMZ+AT101/AT101 (50 µM TMZ, 2.5 µM AT101) for three and six days, respectively. Death rates of primary culture a (PCa, **A**) and primary culture b (PCb, **D**) were obtained by performing a cytotoxicity assay after three and six days of stimulation, respectively (*n* = 3). Live cell imaging using the Polarity Sensitive Indicator of Viability pSIVA^(TM)^ system was performed throughout the treatment period in both mono- and co-cultured GBM cells. The abundance of green (PS) and red (PI) fluorescence was evaluated by images taken every 4 h throughout the treatment period of 6 days (10-fold magnification; PCa: **B**, PCb: **E**). The Gen5 Data Analysis Software (BioTek) was used to quantify the total numbers of PS- and PI-positive cells in relation to the total number of cells within the frame and the ratio of green and red fluorescence per 1000 cells was calculated (PCa: **C**, PCb: **F**). Exemplary data shown; *n* = 3 biological replicates. The significances between different stimulations were determined using a two-way ANOVA test followed by a Tukey’s multiple comparison test (** *p* < 0.01; *** *p* < 0.001). Error bars correspond to the standard deviation.

**Figure 2 ijms-24-09075-f002:**
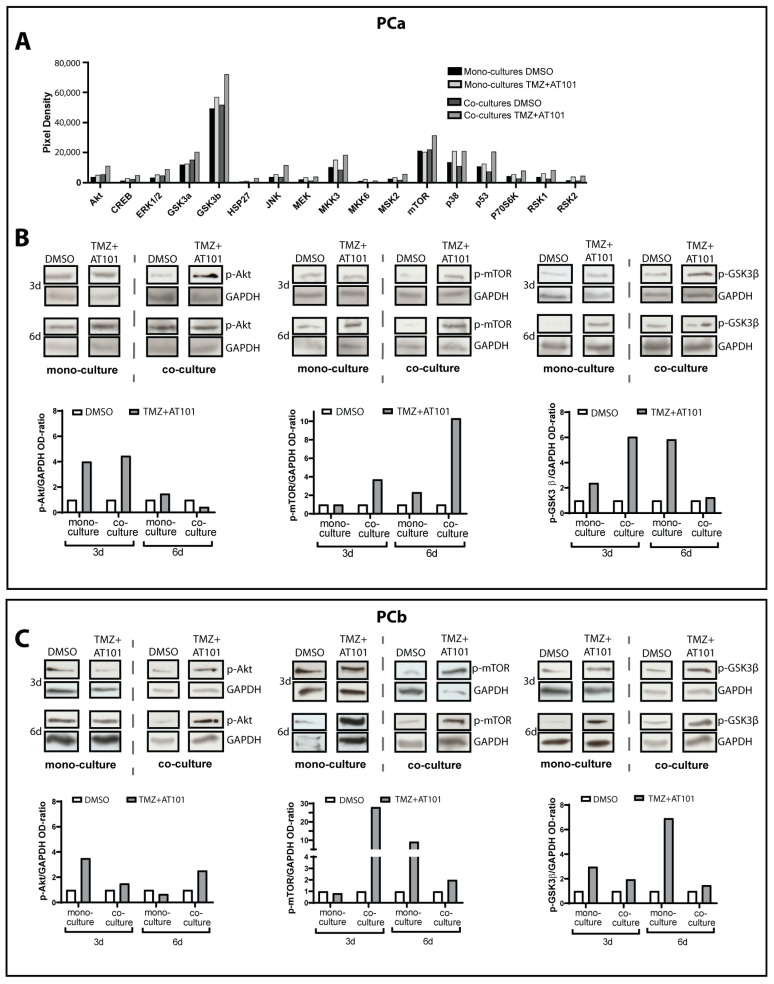
TMZ+AT101/AT101 treatment led to phosphorylation of AKT, mTOR, and GSK3ß in surviving GBM cells. Primary GBM cells were mono-cultured or co-cultured with microglia and astrocytes in defined cellular proportions mimicking an incomplete GBM resection. The cultures were treated with TMZ+AT101/AT101 (50 µM TMZ, 2.5 µM AT101) for three and six days. A human MAPK phosphorylation antibody array of primary culture a (PCa) was performed after 3 days of stimulation (**A**). Western blotting analysis on p-Akt, p-mTOR, and p-GSK3β of mono- and co-cultured primary cultures a (Pca, **B**) and b (PCb, **C**) was performed after stimulation for three and six days, respectively. The obtained p-Akt, p-mTOR, and p-GSK3β signals were normalized to glycerinaldehyde 3-phosphate dehydrogenase (GAPDH) used as loading control. Exemplary data shown; *n* = 2 biological replicates.

**Figure 3 ijms-24-09075-f003:**
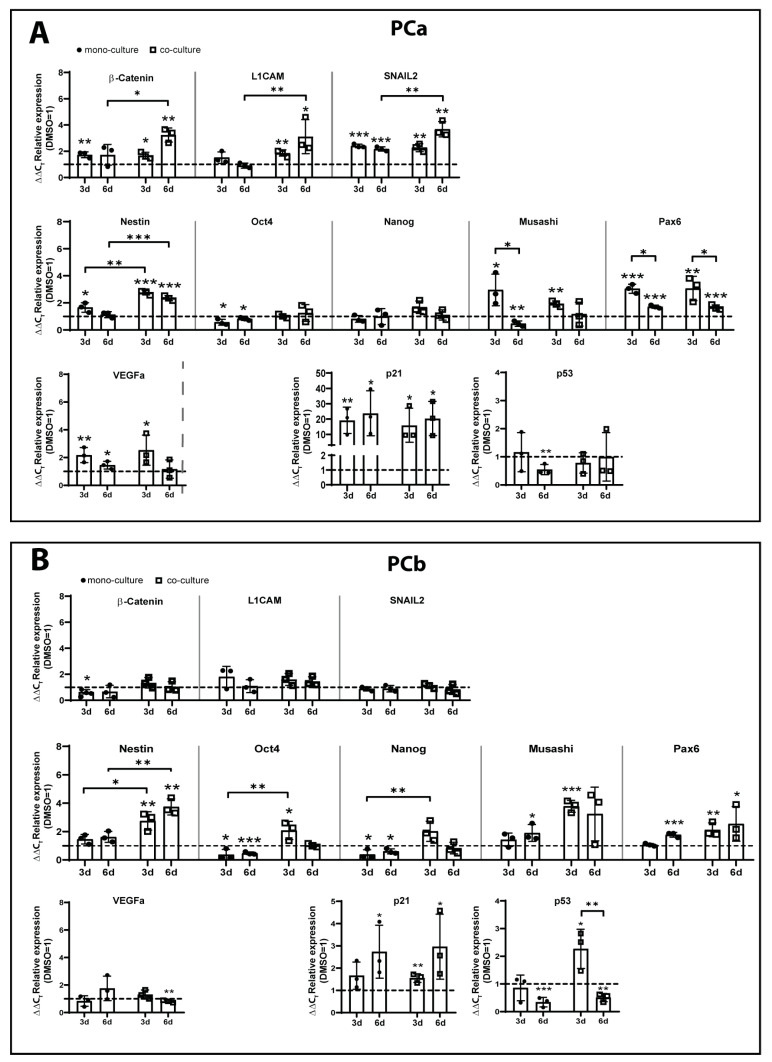
TMZ+AT101/AT101 treatment induced the expression of pro-tumorigenic genes in surviving GBM cells. Primary GBM cells were mono-cultured or co-cultured with microglia and astrocytes in defined cellular proportions mimicking an incomplete GBM resection. The cultures were treated with TMZ+AT101/AT101 (50 µM TMZ, 2.5 µM AT101) for three and six days, respectively. RNA was isolated and qRT-PCR was performed (*n* = 3 biological replicates). Gene expression of different pro-tumorigenic markers was analyzed in primary culture a (PCa, **A**) and primary culture b (PCb, **B**). The induction of gene expression upon stimulation is displayed as n-fold expression changes relative to the DMSO controls after normalization to GAPDH. The significances between the mono- and co-cultures or between different time points were determined by a two-way ANOVA test followed by a Tukey’s multiple comparison test and are indicated on a line linking the bars. Significant differences compared to the DMSO control were determined by a non-paired *t*-test and are indicated directly above the bars (* *p* < 0.05; ** *p* < 0.01; *** *p* < 0.001).

**Figure 4 ijms-24-09075-f004:**
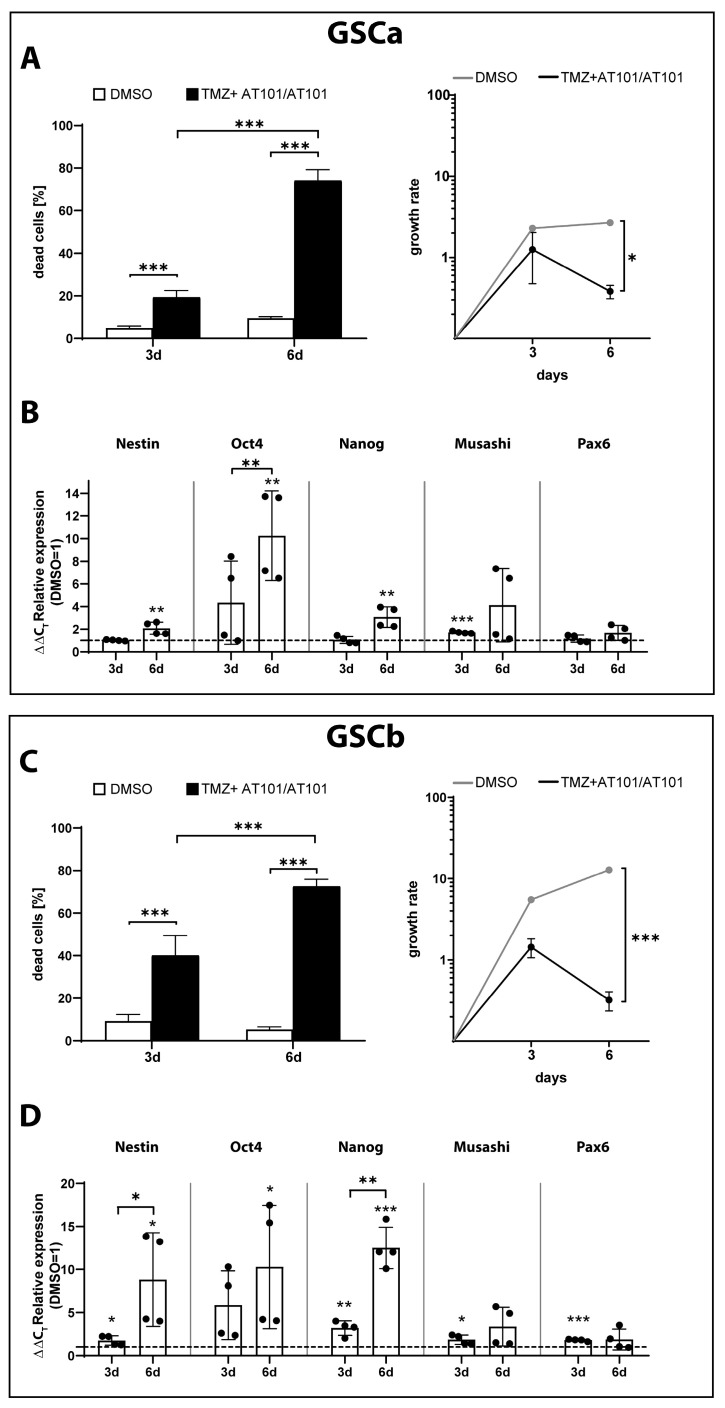
TMZ+AT101/AT101 treatment yielded high cytotoxicity in GSCs and induced the expression of stemness markers in surviving GSCs. Primary GSC cells were mono-cultured and treated with TMZ+AT101/AT101 (50 µM TMZ, 2.5 µM AT101) for three and six days, respectively (*n* = 2 biological replicates with *n* = 2 technical replicates). Death rates of GSC culture a (GSCa, **A**) and GSC culture b (GSCb, **C**) were obtained by performing a cytotoxicity assay after three and six days of stimulation, respectively. RNA was isolated and gene expression of different stemness markers was analyzed in GSC culture a (GSCa, **B**) and GSC culture b (GSCb, **D**) by qRT-PCR. The induction of gene expression upon stimulation is displayed as n-fold expression changes relative to the DMSO controls after normalization to GAPDH. The significances between different time points were determined by a two-way ANOVA test followed by Tukey’s multiple comparison test and are indicated on a line linking the bars. Significant differences compared to the DMSO control were determined by a non-paired *t*-test and are indicated directly above the bars (* *p* < 0.05; ** *p* < 0.01; *** *p* < 0.001).

**Figure 5 ijms-24-09075-f005:**
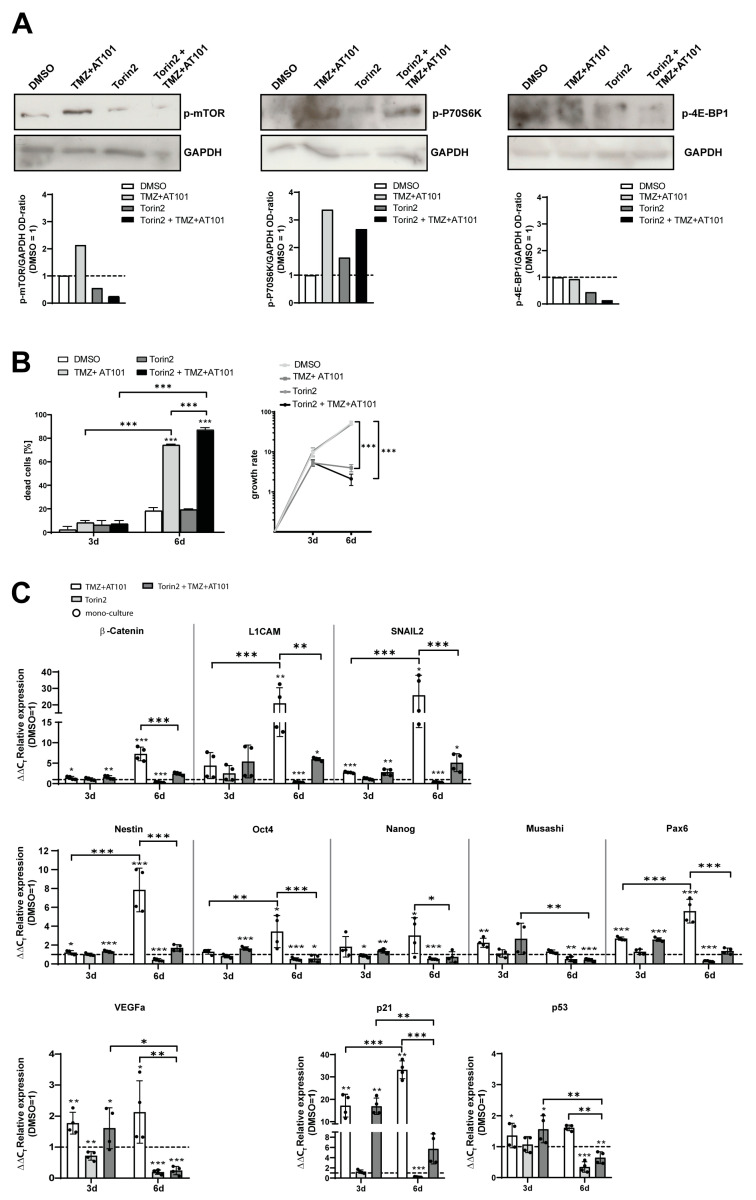
Torin2-mediated mTOR inhibition counteracted TMZ+AT101/AT101-regulated chemoresistance mechanisms of surviving GBM cells. Primary GBM cells were co-cultured with microglia and astrocytes in defined cellular proportions mimicking an incomplete GBM resection. The cultures were treated with TMZ+AT101/AT101 (50 µM TMZ, 2.5 µM AT101), Torin2 (0.5 nM), or a combination of Torin2 and TMZ+AT101/AT101 for six days. Western blotting analysis on p-mTOR, p-P70S6K, and p-4E-BP1 of co-cultured primary cultures a (PCa, **A**) was performed after stimulation for six days, respectively. The obtained signals were normalized to glycerinaldehyde 3-phosphate dehydrogenase (GAPDH) used as loading control. Exemplary data shown; *n* = 2 biological replicates. Death rates of PCa were obtained by performing a cytotoxicity assay after three and six days of stimulation, respectively (*n* = 2 biological replicates with *n* = 2 technical replicates) (**B**). Gene expression of different pro-tumorigenic markers in PCa after three or six days of treatment was determined by qRT-PCR (**C**) (*n* = 2 biological replicates with *n* = 2 technical replicates). The induction of gene expression upon stimulation is displayed as n-fold expression changes relative to the DMSO controls after normalization to GAPDH. The significances between different time points were determined by a two-way ANOVA test followed by Tukey’s multiple comparison test and are indicated on a line linking the bars. Significant differences compared to the DMSO control were determined by a non-paired *t*-test and are indicated directly above the bars (* *p* < 0.05; ** *p* < 0.01; *** *p* < 0.001).

**Figure 6 ijms-24-09075-f006:**
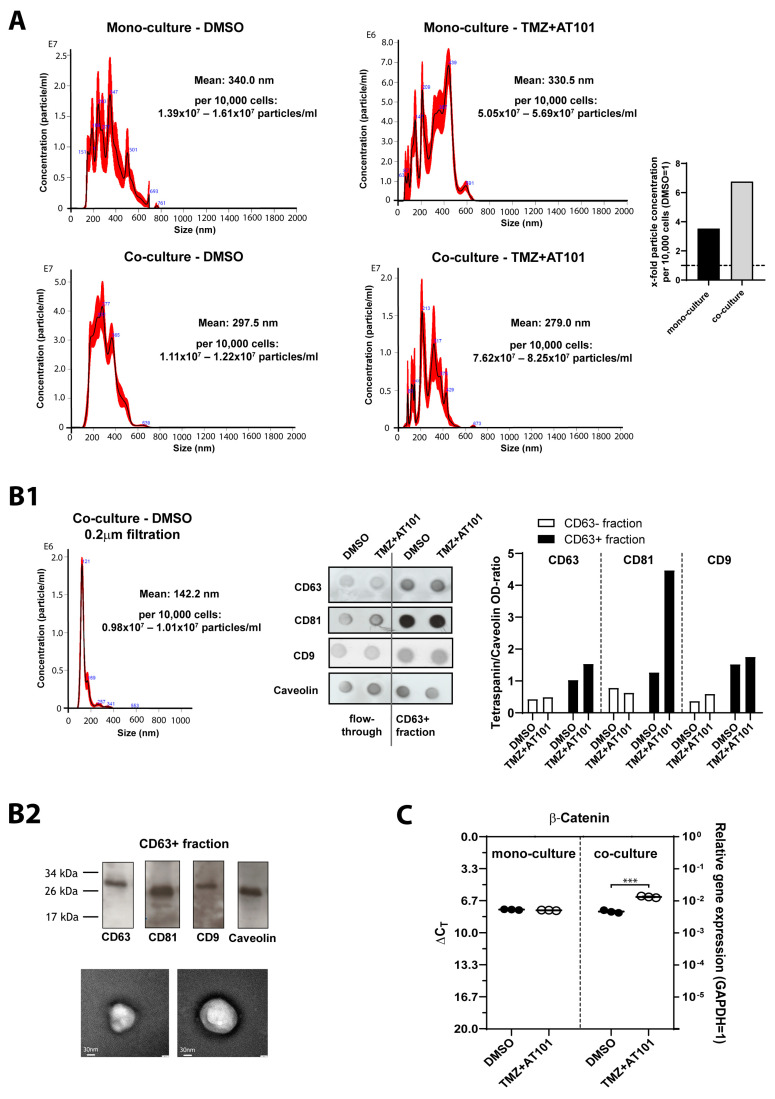
TMZ+AT101/AT101 treatment influenced GBM-derived extracellular vesicles (EVs). Primary culture a (PCa) was mono-cultured or co-cultured with microglia and astrocytes in defined cellular proportions mimicking an incomplete GBM resection. The cultures were treated with TMZ+AT101/AT101 (50 µM TMZ, 2.5 µM AT101) for three days. After three days of stimulation, the media were changed to media supplemented with exosome-depleted fetal bovine serum (FBS), and the cells were incubated for 48 h to allow for the secretion of EVs. The media were harvested and pre-cleared by consecutive centrifugation steps and the EVs were separated by poly(ethylene)glycol (PEG)-mediated precipitation. Obtained EVs of mono- and co-cultured GBM cells (PCa) with and without therapy, respectively, were characterized by nanoparticle tracking analysis (NTA) with respect to their size distribution and quantity (**A**). Exemplary data shown; *n* = 2 biological replicates. Further purification and enrichment for smaller EVs were achieved by filtration and enrichment of CD63+ EVs by magnetic separation (**B1**). Expression of the tetraspanin proteins CD9, CD63, and CD81 was evaluated by Western blotting analysis (**B1**,**B2**). The obtained CD9, CD63, and CD81 signals were normalized to caveolin-1. Morphology of the EVs was assessed by negative staining transmission electron microscopy (**B2**). Gene expression of β-catenin in EVs isolated from PCa after three days of treatment was determined by qRT-PCR (**C**). Significant differences were determined by a two-way ANOVA test followed by a Tukey’s multiple comparison test (*** *p* < 0.001).

**Figure 7 ijms-24-09075-f007:**
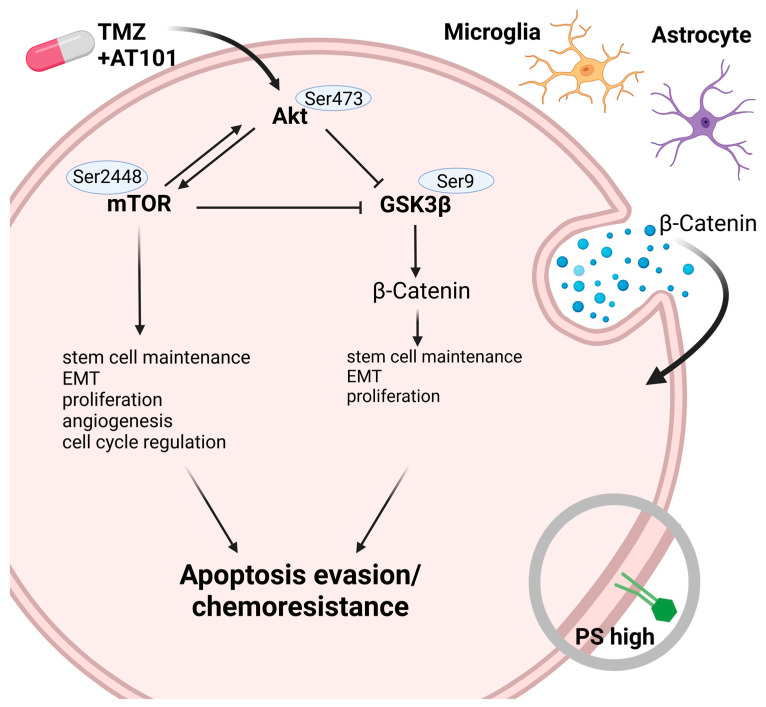
Identified chemoresistance mechanisms of surviving human GBM primary cells after treatment with TMZ+AT101/AT101. TMZ+AT101/AT101 treatment led to an increased number of surviving phosphatidylserine (PS)-positive GBM cells over time, led to phosphorylation of AKT, mTOR, and GSK3ß, resulting in expression of pro-tumorigenic genes, and influenced extracellular vesicles derived from surviving GBM cells.

**Figure 8 ijms-24-09075-f008:**
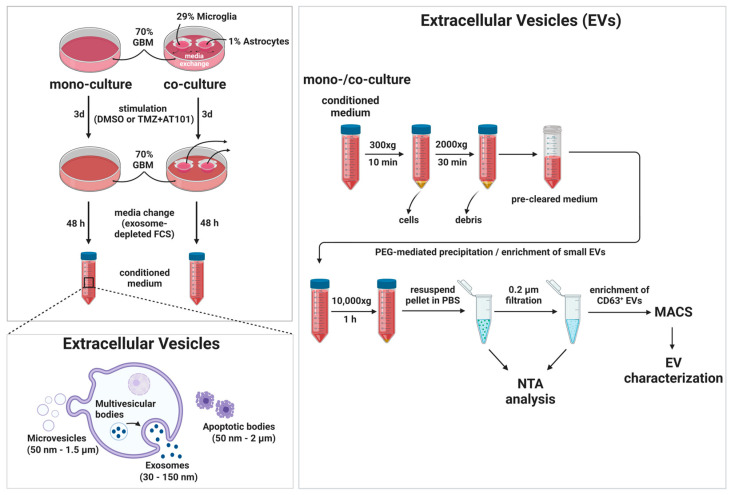
Schematic summary of the procedure for isolation and characterization of extracellular vesicles (EVs). Primary GBM cells were mono-cultured or co-cultured with healthy brain cells in indirect co-culture with cell ratios mimicking the incomplete GBM resection and stimulated with TMZ+AT101/AT101. After three days of stimulation, the media were changed to media supplemented with exosome-depleted fetal bovine serum (FBS), and the cells were incubated for 48 h to allow for the secretion of EVs. The media were harvested and pre-cleared by consecutive centrifugation steps and the EVs were separated by poly(ethylene)glycol (PEG)-mediated precipitation. Further purification and enrichment for smaller EVs were achieved by filtration and enrichment of CD63+ EVs by magnetic separation. Created with BioRender.

## Data Availability

The data presented in this study are available within the article and in Appendix A.

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
