# Peer review of "Sequential Treatment with Temozolomide Plus Naturally Derived AT101 as an Alternative Therapeutic Strategy: Insights into Chemoresistance Mechanisms of Surviving Glioblastoma Cells"

_ijms, 2023, doi:10.3390/ijms24109075_

Round 1

Reviewer 1 Report

See attached file

Reviewer 2 Report

The authors tried an alternative therapeutic strategy using AT101/R(-) enantiomer of the naturally occuring cottonseed-derived gossypol in temozolomide. The authors tried an alternative therapeutic strategy using AT101/R(-) enantiomer of the naturally occuring cottonseed-derived gossypol in temozolomide. The results of the study suggest a possible mechanism of resistance in glioblastoma cells that survive chemotherapy.

The experiments are carefully and meticulously performed and extensively documented, and the text is well written.

There are no particular areas that need reconsideration or revision, but the comprehensive nature of the various experiments makes it difficult to grasp the overall picture.

If possible, it would be even better if the schematic could include a schematic diagram to present the whole picture.

Round 2

Reviewer 1 Report

The revised manuscript by Hellmold et.al includes additional data which is informative and the new text addressing the concerns raised in the previous submission provide greater clarity and highlight some of the limitations of the study which should prove valuable and useful to readers.